# Automated Parameter Extraction for Biologically Realistic Neural Networks: An Initial Exploration with Large Language Models

## Abstract

In computational neuroscience, extracting parameters for constructing biologically realistic neural models is a resource-intensive task that requires continuous updates as new research emerges. This paper explores utilizing large language models (LLMs) in automating parameter extraction from scientific literature for biologically realistic neural models. We utilized open-source LLMs via Ollama to construct KGs, capturing parameters such as neuron morphology, synapse dynamics, and receptor properties. SNNBuilder Gutierrez et al. (2022), a framework for building spiking neural network (SNN) models, serves as a key validation example for our framework. However, the methodology we outline here can extend beyond SNNs and could applied to systematic modelling of the brain.By experimenting with different prompting strategies—general extraction, in-context hints, and masked prompting—we evaluated the ability of LLMs to autonomously extract relevant data and organize it within an expert-base or data-driven ontology, as well as to infer missing information for neural model construction. Additionally, we implemented retrieval-augmented generation (RAG) via LangChain to further improve the accuracy of parameter extraction through leveraging external knowledge sources. Analysis of the the generated KGs, demonstrated that LLMs, when guided by targeted prompts, can enhance the data-to-model process, paving the way for more efficient parameter extraction and model construction in computational neuroscience.

## 1 Introduction

In computational neuroscience, parameterization of brain models is a time-consuming task. As it requires identifying parameters related to the structure and function of the brain region being modeled. Data-to-model frameworks, such as SNNbuilder (Spiking Neural Network builder) Gutierrez et al. (2022), can assist in model building; however, since these parameters primarily come from scientific publications, their extraction requires continuous updates as new research emerges, making the process both time and resource intensive.

LLMs (Large Language Models) can enhance the data-to-model process in computational neuroscience and have shown promise in certain generalization tasks out of their domain Yang et al. (2024). LLMs provide a valuable methlodgy for exploring vast amounts of information structred and unstructed information through their large paramter counts and pretrained weights.

However, their full potential in research remains largely untapped without agumenting external data. By augmenting LLMs with external data in a process known as retrieval augmented generation (RAG) Lewis et al. (2020), model performance in NLP tasks can be greatly improved. In neuroscience researchers could have access to a more intuative approach to identify paramters by utilizing scientific publications and interacting via natural language with the model and database.

Furthermore, extracted brain data can be structured to mirror the brain's architecture. Experts could develop a brain ontology to organize this information intuitively and in an easily understandable format. State-of-the-art research for creating graph structures based on scientific text, such as Graph RAG Edge et al.

(2024), can often generate structures based on their probabilistic reasoning rather than expert-driven design, which results in less interpretable outcomes.

The degree to which these LLM-generated structures align with expert-based ontologies remains largely unexplored. In this work, we carried out experiments with different promoting strategies to guide the graph construction process towards a more comprehensive model.

## 2 RELATED WORK

Recent work on ontology-guided knowledge graph (KG) construction Cauter & Yakovets (2024) has demonstrated the effectiveness of LLMs such as Llama-2 and Llama-3 in extracting domain-specific facts. Using a few semantically similar examples, the researchers could compare their performance to state-of-the-art fine-tuning methods on the Llama-3-70B-Instruct model. This approach aligns with this research, with LLMs used to extract SNN parameters from scientific literature and relying on similar prompting techniques.

Our work aligns with the broader trends we have seen in AI in healthcare, where LLMs can aid in parsing large amounts of unstructured medical data Kather et al. (2024). Recent work with LLMs in clinical health extraction contexts has shown improvement when using in-context learning approaches and external knowledge bases Li et al. (2024). Such progress highlights the trend in AI in health domains toward identifying valuable data points from large volumes of unstructured texts, thus reducing the human need to adhere to strictly structured formats, which is typical for electronic health records Nashwan & AbuJaber (2023).

## 3 PRELIMINARIES

Early experimentation in this research explored using RAG with baseline proprietary models such as GPT-4o based on the GPT-4 architecture OpenAI (2024) to automate the extraction of neural parameters from the scientific literature.

We implemented a one-shot promoting approach to extract key neural parameters that match specific fields within a KG. The graph was based on a hand-authored ontology with a predefined structure, as seen in Figure 1. The graph aimed to represent brain circuits and their components, including species, brain regions, neurons, and their connections. The graph structure was provided to the LLM as Cypher, the query language used by the open-source graph database Neo4j. This prompting approach of providing a base hand-authored graph to augment additional nodes has been shown to be effective in prior research Jhajj et al. (2024).

We carried out a baseline prompting approach using the GPT-4o. An example of this approach with a truncated prompt and response can be seen in Table 1.

This baseline RAG implementation was carried out using LangChain [1], and it did provide useful insights outside of pronpt. For example, the LLM could suggest other similar parameters that were outside of the ontology we had provided. In Table 1, for example, we can see the model replied with the neuron parameter of "Firing Rate" when that was not provided in the initial prompt.

---

[1]https://www.langchain.com/

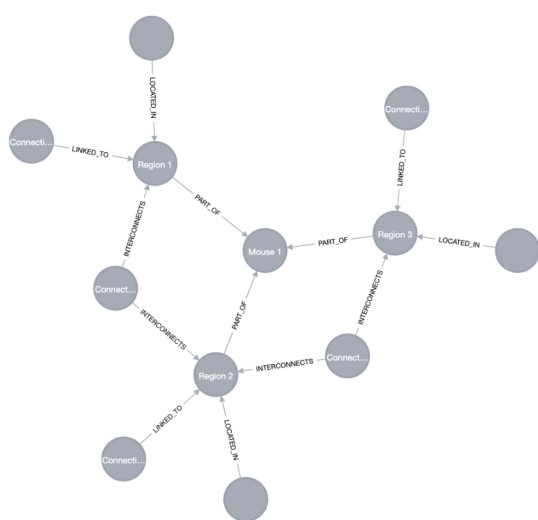

Figure 1: Handcrafted Ontology for Brain Modelling

Table 1: Truncated prompt and response for neural parameter extraction

| Prompt | Response |
| --- | --- |
| Please go through the attached paper and extract parameters that match or are similar to the following fields: [list of parameters]. So they can be augmented to this KG. | I extracted the relevant information from the paper to match your specified fields: neuron types, firing rates, and numbers/morphology. Here are the details: |
| `CREATE (A:BrainCircuit {name: 'Brain Circuit'})`
`CREATE (A1:Species {name: 'Species'})`
`CREATE (A2:BrainRegions {name: 'Brain Regions'})`
`CREATE (A3:Neurons {name: 'Neurons'})`
`CREATE (A4:Connections {name: 'Connections'})`
`CREATE (A5:Inputs {name: 'Inputs'})` | Neuron Types:
- Medium Spiny Neurons (MSN)
- Fast-Spiking Interneurons (FSI)
- Subthalamic Nucleus Neurons (STN)
- Globus Pallidus externus Neurons (GPe)
- Globus Pallidus internus/Substantia Nigra reticulata Neurons (GPi/SNr)
- Corticostriatal Neurons (CSN)
- Pyramidal Tract Neurons (PTN)
- Centromedian/Parafascicular Thalamic Nucleus Neurons (CM/Pf)
Firing Rates:
MSN: 300 Hz
STN: 300 Hz
GPe: 400 Hz
GPi/SNr: 400 Hz |

Later experimentation shifted towards using Graph RAG Edge et al. (2024) to aid in KG generation. Graph RAG is an approach to RAG that creates a graph-based index, which is later queried to respond Edge et al. (2024). Graph RAG utilizes KGs, allowing for a deeper understanding of datasets and providing richer responses as demonstrated in Larson & Truitt (2024). Recent has shown LLMs are effective in triplet extraction for KG construction Papaluca et al. (2024); Zhang et al. (2024)

### 3.1 Hand Authored Ontology

A detailed hierarchical ontology guided the model's understanding of neural structures. The ontology can be seen following this:

- **Neuron**
  - Neuron name
  - Number of neurons (depends on the species)
  - **Dendrite**
    - \* Morphology
      - · Diameter
      - · Spatial domain (length, mean length, size of the dendritic field, spatial distribution, extent, spread)
      - · Spine density
  - **Axon**
    - \* Topology
      - · Boutons count (number of boutons)
      - · Spatial domain (length, mean length, size of the axonal arbor, spatial distribution, extent, spread)
  - **Synapse**
    - \* Synaptic delay
    - \* Neurotransmitter release
  - **Receptor**
    - \* Receptor type
    - \* Neurotransmitter related
    - \* Receptor spatial location (distance to the soma)
    - \* **Dynamics**
      - · PSPs (post-synaptic-potential amplitude, rise time)
      - · Plasticity (rules, dynamics)
  - **Electrophysiology**
    - \* Firing rates (at resting, during activity, during disease, etc.)
    - \* Membrane dynamics (resting potential, membrane potential, capacitance, resistance, time constant, refractory period, spike threshold, reset potential)

When provided with this detailed, hierarchical structure, the potential of the Graph RAG approach to generate a KG that preserves these higher-order relationships between the entities was a key focus of our research. For this, we used a similar prompt to that in Table 1 but with a different ontology with LLama-3.1-70B Dubey et al. (2024). However, as shown in Figure 2, the resulting KG failed to maintain the intended hierarchy. This is an important issue to note as ontology is critical to understanding the complex relationships that are present in the brain, such as between neurons, synapses, and receptions. Structurally, this loss is a critical limitation in this approach, adversely impacting the interpretability and utility of the created graph in reflecting the complexity of neural structure and function.

This result suggests that the LLM with the Graph RAG KG generation approach can struggle to maintain the intended multi-level hierarchy of the ontology when generating the graph, mainly when using the Base Prompt alone.

## 4 Methodology

### 4.1 Knowledge Graph Construction

Our approach focused on creating KGs by extracting structured information from scientific literature using LLMs. For this, open-source models such as Llama3.1 Dubey et al. (2024) via Ollama [2] to construct KGs representing the content within

---

[2] https://github.com/ollama/ollama

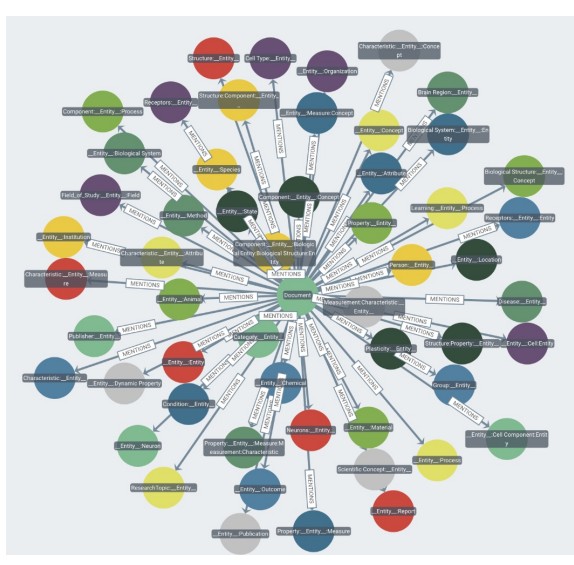

Figure 2: Base graph rag generated graph

the scientific texts and to support the parameterization of the architecture of spiking neural networks (SNNs). We identified key attributes such as neuron morphology, synapse dynamics, and receptor properties using different prompting strategies. The LLM outputs were parsed into nodes and edges, which were subsequently visualized as KGs and stored in Neo4j, a graph database. This allows for unstructured biological data to be formatted into coherent structures that can be analyzed and compared.

This work only utilized open-source models as proprietary LLMs such as GPT-4 OpenAI (2024) often lack reproducibility due to their black box nature and constant updates Ollion et al. (2024); Ferrari et al. (2023). Besides that, the cloud-hosted nature of many proprietary models and the deprecation of older models may hinder reproducibility. There is ongoing concern surrounding reproducibility for proprietary and open-source models as new architectures are released Vaugrante et al. (2024). However, leveraging open-source models gives users greater control and understanding of model usage and performance Ollion et al. (2024). It can allow researchers to maintain their implementations without concerns of cloud-based depreciation, over which they have minimal control.

## 4.2 EXPERIMENTAL SETUP

To assess the performance of different LLMs in extracting relevant SNN parameters, we trialed three different prompting strategies.

### 4.2.1 EXPERIMENT 1: BASE EXTRACTION

We used a general prompting strategy with minimal guidance in the first experiment. The goal was to evaluate the LLM's ability to extract key SNN parameters autonomously without domain-specific hints. The following prompt was used:

"Extract spiking neural network parameters from the following scientific text:"

Through this first experiment, the LLM was expected to identify critical SNN-related information, such as neuron morphology, synaptic delay, and receptor properties, based solely on its pre-trained understanding of scientific texts and the provided scientific article. These results provided a baseline for the LLM's ability to extract relevant information with little external knowledge. The resulting parameters for this KG can be seen in 3a. The goal was to see nodes representing entities such as neurons and synapses and edges representing relationships between them

### 4.2.2 EXPERIMENT 2: IN-CONTEXT HINTS

In the second experiment, we used in-context hints within the prompt. This would guide the LLM in focusing on specific SNN parameters. The hint was based on the prior ontology in 3.1.

> "You are tasked with extracting important parameters for building spiking neural networks (SNNs). Focus on parameters such as neuron morphology, dendrite structure, synapse delay, receptor types, and electrophysiological properties.

This experiment was done to assess the impact of domain specific hints on the LLMs ability to generate more comphrensive responses when prompted for accurate parameters. The resulting KG can be see in 3b.

### 4.2.3 EXPERIMENT 3: MASKED PROMPTING

This experiment used a masked prompting strategy where the model was only provided with a partial prompt with different parts of the ontology from 3.1. This was done to evaluate the LLM's ability to determine entities and relationships not stated in the input prompt. The prompt we used was:

> You are tasked with extracting parameters for building spiking neural networks (SNNs). Focus on - Neuron Name - Dendrite (Morphology) - Axon (Topology, Spatial domain) Try to infer missing details related to synapse dynamics, receptor types, and membrane properties."

The LLM, with this prompt, was guided to infer connections between entities even if there was a lack of information in the text. The resulting KG can be seen in 3c.

## 5 EXPERIMENTS AND RESULTS

The KGs generated from all three experiments were evaluated using several graph-based metrics to assess their quality and structure. The results in Table 2 and 3 were created from a small corpus of neuroscience papers. The resulting set of KGs described by the metrics in Table 2 can be seen in Figure 3.

Table 2: Prompting results

| Prompt Type | Nodes | Edges | Leiden Modularity |
|---|---|---|---|
| Base Prompt | 144 | 141 | 0.63 |
| In-context Prompt | 325 | 422 | 0.60 |
| Masked Prompt | 326 | 360 | 0.55 |

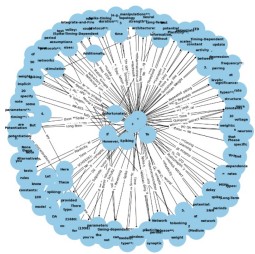 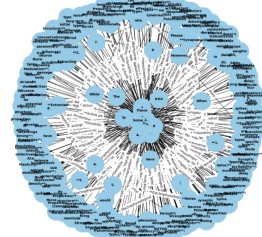 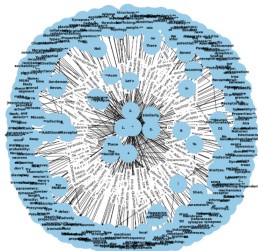

(a) Base Prompt Graph   (b) In-Context Prompt Graph   (c) Masked Prompt Graph

Figure 3: Comparison of knowledge graphs generated using different prompting methods

Table 3: Average degree centrality results for different prompting methods

| Experiment | Node Count | Edge Count | Average Degree Centrality |
|---|---|---|---|
| Base Prompt | 66 | 61 | 0.0284 |
| In-Context Prompt | 370 | 379 | 0.0055 |
| Masked Prompting | 386 | 447 | 0.0060 |

## 6 DISCUSSION

Table 2 contains the results for experiments 4.2.1, 4.2.2, and 4.2.3. For these experiments, we used Leiden Modularity Traag et al. (2019) to assess the community structure within the KGs. A higher modularity score indicates a more well-defined cluster of entities.

Despite producing the fewest nodes, 144, and edges, 141, the base prompt achieved the highest Leiden Modularity score of 0.63. This suggests that while fewer nodes and entities were created with this method, they were clustered into more well-defined communities. It also indicates that there are more intra-cluster relationships.

The in-context prompt extracted a significantly larger number of nodes at 325 and edges at 422 but saw a slight reduction in Leiden Modularity. While the LLM did extract more entries and relationships, the lower score suggests that additional noise may have been introduced through these new connections. However, we still see a well-defined community structure via a modality of 0.60, suggesting that the graph's coherence was not severely compromised when compared to the base graph.

The masked prompt did produce the largest graph with 326 nodes but had slightly fewer edges at 360. It also had the lowest modularity score of 0.55. This suggests that while it did capture a comparable amount of entities to the in-context approach, they were not as well connected and formed and had more diffuse clusters. This could entail that this approach suggests inferred relationships that may not strongly correlate, resulting in a lower modularity score of 0.55.

Using another small corpus of neuroscience papers, we tested the Average Degree Centrality, which is used to evaluate the connectivity within each graph. The results for this can be seen in Table 3

The formula for Average Degree Centrality is given by:

$$\text{Average Degree Centrality} = \frac{1}{N} \sum_{i=1}^{N} C_D(v_i)$$

where $N$ represents the total number of nodes, and $C_D(v_i)$ is the degree centrality of node $v_i$. This metric provides insight into how densely connected the graph is, with higher values indicating more connections per node on average.

When looking at the prompting approahces we can see that the base prompt generated a smaller, denser graph with the highest average degree centrality 0.0284, this shows that extracted entities were more connected. However, for these results it should be noted that the generaated graph for the base prompt was much smaller the other two prompting approaches.

Compared to this the in-context prompt nad masked prompt made a largber graph structure but had lower average degree centrality values of 0.0055 and 0.0060 respectivly. This shows a broader but more sparse network of relationships.

The current approach to KG generation using Graph RAG faces several limitations that impact the quality and accuracy of the generated graphs. The small corpus size, which was only selected to be a few papers, would affect how representative the graphs are. A smaller corpus will not be as generalizable to the entire domain, resulting in less robust graphs. We also currently lack a mechanism to validate our nodes. Currently, Graph RAG can incorporate large amounts of text as it is good at modeling domain knowledge, but for our use case, it struggles at only identifying

and extracting key entities for brain modeling. This can make the overall graph structure more nosy and include poorly correlated information.

## 7 CONCLUSION

As a first step in exploring the use of Graph RAG for modeling brain-related knowledge graphs, we generated graphs based on a limited corpus of neuroscience papers. These initial experiments helped understand how KGs were by using LLMs and the impact of prompting approaches on their structures. However, several significant challenges remain to ensure its alignment with real-world brain modeling, and currently, the results of our experiments do not fully capture the complexities of modeling the brain; the current graphs may not fully capture the structure of the brain and various organizations.

Future work can focus on two main areas to address the aforementioned limitations. The first is to use a larger corpus of papers to represent more of the complexity of neuroscience and brain modeling. Second, we can use a node validation step to ensure that only relevant entities based on ontology are present in the graph. Techniques such as prompt tuning and finetuned models can aid in achieving this. Additionally, the creation of a neuroscience-based QA dataset to validate neuronal parameters can help evaluate our generated KGs.

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
