# OpenReview forum: "Automated Parameter Extraction for Biologically Realistic Neural Networks: An Initial Exploration with Large Language Models"
_ICLR.cc/2025/Conference — Submitted to ICLR 2025_

### Official Review · Reviewer_XeV2 · 2024-10-15

**Soundness:** 1
**Presentation:** 1
**Contribution:** 1
**Rating:** 1
**Confidence:** 5

**Summary:**

This paper explores the feasibility of using large language models (LLMs) to automatically extract the parameters required to build biologically realistic neural networks from scientific literature. The authors evaluated the ability of LLMs to extract parameters such as neuronal morphology, synaptic dynamics, and receptor properties through different prompting strategies (general extraction, contextual prompts, and masked prompts), and used retrieval-augmented generation (RAG) technology to improve the extraction accuracy. The main verification case of the study is the SNNBuilder framework, which is used to build spiking neural networks (SNNs). The results show that LLMs can effectively promote data extraction and organization in neural model construction under specific prompts, providing an efficient and automated parameter extraction method.

**Strengths:**

The author focuses on two core areas in the paper:
1. Using LLM to automatically build and update knowledge graphs from unstructured data
2. Organizing complex experimental data in neuroscience and converting them into computable models

I think both areas are very interesting and have practical application value.

For 1, knowledge graphs are difficult to be expanded to a larger scale, updated and deployed in real time due to their high construction and maintenance costs. However, graph databases provide a reliable way to store and reason about complex information. On the one hand, knowledge graphs can benefit from LLM's powerful natural language parsing capabilities, and on the other hand, LLM can benefit from knowledge graphs' reliable information storage methods to alleviate hallucinations.

For 2, there is a large GAP between current AI research and neuroscience research: researchers in the two fields find it difficult to communicate using the same academic language. For example, a large number of experimental results produced in neuroscience research are difficult to be converted into clear and reusable computational models in the field of machine learning; on the other hand, advanced machine learning models (such as LLM, LVM) lack biological interpretability, and it is difficult to find corresponding experimental evidence in neuroscience. Bridging this GAP in any way will help the two fields communicate and benefit each other.

**Weaknesses:**

As I mentioned in the strengths section, this paper focuses on interesting problems. But I think it is a simple technical report rather than an academic paper. It is not appropriate to publish it at ICLR. (Btw I think this paper uses the iclr template incorrectly, and the margins of the main text are incorrect.)

The specific areas that can be improved are as follows:
1. Lack of novelty in the method: The author did not do any further work except proposing three prompts.
2. Lack of experiments: There are many technical details in designing computational models in the field of neuroscience using LLM. The authors did not conduct/show these experiments.

I will elaborate on these weaknesses in the Question.

**Questions:**

1.Does the SNN model (entity, relation, entity triple) extracted using LLM strictly follow the prompt? Does the author's method in Section 4 effectively alleviate the problems pointed out in Section 3? :"This result suggests that the LLM with the Graph RAG KG generation approach
can struggle to maintain the intended multi-level hierarchy of the ontology when generating the graph, mainly when using the Base Prompt alone."

2.How to measure the quality of knowledge graph extracted by LLM? According to Figure 3, there are a large number of meaningless entities in the generated graph, such as "while, to, the, here, i". Such a graph is far from meeting the standards of practical applications.

3.An important problem of knowledge graph is entity disambiguation, but it is not shown in the author's research. The author used the well-known open source framework GraphRAG/Neo4j. It is obvious that the existing methods are not good enough, but the author did not make more contributions.

4.Are the SNN models generated by LLM biologically correct and meaningful to biology? For the machine learning community, can these models be used or even lead to better machine learning models? This paper lacks any measurement of the effectiveness of these models.

---

### Official Review · Reviewer_rbRP · 2024-10-19

**Soundness:** 1
**Presentation:** 1
**Contribution:** 1
**Rating:** 1
**Confidence:** 3

**Summary:**

This work documents an initial exploration of the use of LLMs (such as GPT4o and LLama3.1) in conjunction with different prompting strategies (such as RAG) to automatize the extraction of computational neuroscience relevant parameters (in particular for SNN) into Knowledge Graphs (KG) from relevant literature.

**Strengths:**

- The project aims to tackle relevant research questions.
- The use of open source LLMs ensures wider applicability of the method.
- The conclusions does acknowledge some limitations of the work.

**Weaknesses:**

* 1\. The main claims/contributions in the abstract are hard to relate to what experiments and results are reported in the paper:
   - the “accuracy of parameter extraction” was not evaluated?
   - the “data-to-model” process was not evaluated?
   - the “SNNBuilder” was not used in any evaluation?
   - it's unclear how the characterization of the graphs using the  Leiden Modularity relates to any of the claims
   - => the suitability of the generated graphs for comp. neuroscience model generation/configuration was therefore neither qualitatively nor quantitatively shown anywhere
* 2\. Many methodological details are unclear and missing, and therefore it is unclear how to interpret the results, or how to reproduce the results? (see Questions).
* 3\. Many formulations are hard to properly understand, as they are presumably imprecise or incorrectly using technical terms, among others:
   - “... promoting strategies …” => prompting strategies? (line 58-59)
   - “... one-shot promoting …” => prompting? (line 82-83)
   - Spelling errors “nda“ and “sparge“?  (line 169-171)
   - Unclear what is meant by “ parameterization of brain models” here? (line 34)
   - Unclear what is meant by “agumenting external data” here? (line 44-45)
   - Unclear what “brain data” is referring to? (line 51)
   - “Structurally, this loss is a critical limitation in this approach,…” => the loss of structure? (line 202-203)
* 4\. The structuring of the paper is confusing and hard to follow:
   - The “Preliminaries” section mostly describes experiments and methods unrelated to the rest of the paper, and presumably serves as explanation as to why a different approach is introduced later. Possibly something for the Appendix?
   - Some results are first described in the Discussion, and additional experiments are introduced in the Discussion.
* 5\. Figure 3 is mostly unreadable, and readable parts mostly reflect non neuroscience related terms:
   - E.g. 3.c: “Keep”, “To”, “It”, “Assuming”, “Based”, “Name:”, “Not”, “\”, “don’t”, …
* 6\. The related literatur section is missing methodologically more closely realted papers.

**Questions:**

1) Which paper was used as source for Table 1?
2) How was the hand authored ontology in section 3.1 derived?
3) How are the Figure 1 and the Ontology from 3.1 related?
4) What are the exact equations for “Leiden Modularity“?
5) What for / where was the “SNNBuilder” actually used?
6) What are the “different parts of the ontology from 3.1”, and were they actually used? (line 286)
7) What is the corpus of papers? (line 301)
8) What other corpus of papers? (line 351)
9) How does evaluating “Leiden Modularity” support the claims in the abstract?
10) How does evaluating “Average Degree Centrality” support the claims in the abstract?

---

### Official Review · Reviewer_C942 · 2024-11-03

**Soundness:** 1
**Presentation:** 1
**Contribution:** 1
**Rating:** 3
**Confidence:** 4

**Summary:**

This paper investigates the use of open-source LLMs to automatically extract parameters from scientific literature for building biologically realistic neural models, particularly SNNs. By employing various prompting strategies and retrieval-augmented generation, the authors successfully constructed knowledge graphs that capture essential neural attributes, enhancing the efficiency of model parameterization.

**Strengths:**

1. The intuition is good.

**Weaknesses:**

1. Formatting Issues: The paper's format is problematic, with the main text appearing excessively narrow, which does not adhere to standard formatting guidelines.

2. Insufficient Detail: The paper is notably short, providing an overly simplistic description of the methodologies employed. Additionally, the experimental section is minimal, lacking comprehensive analysis and validation.

3. Project-Oriented: The content resembles a preliminary project report rather than a fully developed research paper. It falls significantly below the standards expected for publication in venues like ICLR, both in terms of depth and rigor.

**Questions:**

1. Could you provide a more detailed description of your method?

2. Could you give more experimental results?

---

### Official Review · Reviewer_pv7Z · 2024-11-03

**Soundness:** 2
**Presentation:** 2
**Contribution:** 2
**Rating:** 5
**Confidence:** 4

**Summary:**

This paper investigates the use of large language models (LLMs) to automate parameter extraction from scientific literature for constructing biologically realistic neural models, capturing details like neuron morphology, synapse dynamics, and receptor properties.

**Strengths:**

***1.*** Using LLMs to extract parameters for building SNNs is an interesting topic.

**Weaknesses:**

***1.*** The paper lacks more experimental results and does not provide a detailed analysis of the findings.

***2.*** The paper has an unusual format and does not appear to follow the original template.

***3.*** The phrasing throughout the paper is unclear.

***4.*** The authors need to conduct extensive restructuring and experimental analysis.

**Questions:**

Please refer to the "Weaknesses" section for further details.

---

### Meta-Review · Area_Chair_kX2T · 2024-12-22

**Metareview:**

This paper introduces a method to automatically extract biologically realistic neural model parameters from scientific literature using large language models (LLMs). The authors are motivated by the problem that traditional parameter extraction tasks are resource-intensive and require frequent updates. The authors evaluate the ability of LLMs to extract parameters such as neuronal morphology and synaptic dynamics through prompting and retrieval-augmented generation (RAG) techniques, and construct a knowledge graph. Experimental results show that LLMs can effectively extract and organize relevant data under specific prompts, providing a new method for efficient and automated neural model parameterization and construction.

After the initial reviewing stage, 4 reviewers rate 1, 1, 3, and 5, respectively. Most reviewers agree that the theme of this paper, which uses LLMs to extract neuronal parameters to build neuroscience models, is innovative, which is the main advantage of this method. However, there are also some criticisms. Reviewers pv7Z and C942 believe that the paper lacks experimental results and detailed analysis, and has problems with format and expression, and is more like a preliminary technical report overall. Reviewers rbRP and XeV2 pointed out that the method lacks novelty, the experimental design is insufficient, and its practical application value has not been verified. In addition, the logic of the paper is not clear enough. I think these criticisms are reasonable. I suggest that the author strengthen experimental verification, improve the quality and biological significance of the knowledge map, improve the structure and expression of the paper, and supplement the detailed analysis of the methods and results to enhance the academic depth and rigor.

Since the score of this paper did not increase after the rebuttal period, I think this paper has not reached the level that can be accepted. Therefore, it was finally decided to reject this paper.

**Additional Comments On Reviewer Discussion:**

The author did not discuss the reviewers' suggestions during the discussion stage.

---

### Decision · Program_Chairs · 2025-01-22

Reject